# Dietary Silk Peptide Inhibits LPS-Induced Inflammatory Responses by Modulating Toll-Like Receptor 4 (TLR4) Signaling

**DOI:** 10.3390/biom10050771

**Published:** 2020-05-15

**Authors:** Sungwoo Chei, Hyun-Ji Oh, Kippeum Lee, Heegu Jin, Jeong-Yong Lee, Boo-Yong Lee

**Affiliations:** 1Department of Biomedical Sciences, CHA University, Gyeonggi-do 13488, Korea; sungwoochei@gmail.com (S.C.); guswl264@naver.com (H.-J.O.); joy4917@hanmail.net (K.L.); heegu94@hanmail.net (H.J.); 2Worldway Co., Ltd., Sejong 30003, Korea; dalgoozi@hanmail.net

**Keywords:** silk peptide, inflammatory response, TLR4 signaling, lipopolysaccharide, cytokine, bone-marrow derived macrophage (BMDM)

## Abstract

Acid-hydrolyzed silk peptide (SP) is a valuable material that has been used traditionally to treat various diseases, however, the mechanism by which it affects inflammatory responses is unknown. To examine the effects of SP on inflammatory responses, we used macrophages as a vehicle for examining signaling via toll-like receptor 4 (TLR4), which plays an important role in innate immune responses to pathogenic infections and pathogen-derived molecules such as lipopolysaccharide (LPS). We then confirmed the anti-inflammatory effects of SP by examining lymph node, spleen, and serum samples from C57BL/6 mice injected with LPS. We also used LPS-induced bone marrow-derived macrophages and RAW264.7 cells (a murine macrophage cell line) to identify the mechanism by which SP modulates immune responses via the TLR4 signaling pathway. In addition, we showed that SP prevents LPS-induced production of nitric oxide and reactive oxygen species. In summary, SP inhibits LPS-induced inflammatory responses by modulating the TLR4 signaling pathway.

## 1. Introduction

Acid-hydrolyzed silk peptide (SP) derived from *Bombyx mori* cocoons is a highly valuable material that historically has been used to treat various conditions [1,2,3]. The SP biopolymer is produced and used by silkworms to form cocoons, which provide a protective environment during metamorphosis [4]. Traditionally, SP (which comprises various biomolecules) is used pharmacologically in the form of a powder or as an extract, and is considered a functional food in Asian countries [4,5] as it has anti-obesity and anti-cancer effects, however, it has also been studied on a specific anti-inflammatory pathway [6].

At the early stage of infection, the innate immune system produces a rapid inflammatory response, which inhibits the growth and spread of infectious pathogens [7,8]. The innate response also triggers an adaptive immune response, which eliminates the pathogen [9,10]. The innate immune system, which is mediated mainly by macrophages and dendritic cells, is triggered by a limited number of germline-encoded pattern recognition receptors (PRRs) [11]. These receptors have evolved to recognize conserved molecular patterns expressed by pathogens, so-called pathogen-associated molecular patterns (PAMPs). These include lipopolysaccharide (LPS) [7,9,11,12,13]. Typically, LPS comprises a hydrophobic domain known as lipid A, an oligosaccharide, and a polysaccharide [14]. LPS is a common PAMP found in the outer membrane of Gram-negative bacteria and it binds to a protein family (toll-like receptors (TLRs)) expressed on the surface of immune cells [15].

TLRs are type I integral receptors comprising an intra- and an extracellular domain and a single transmembrane helix [16]. Research on *Drosophila* receptors led to the finding that mammalian TLRs also function as PRRs. TLR4 is crucial for immune recognition of Gram-negative bacteria, making it a crucial component of defense against pathogens [9]. Activation of innate immune system via TLR4 triggers adaptive immune responses. TLR4 and its co-receptor myeloid differentiation factor 2 (MD-2) recognize diverse LPS molecules [17]. After binding LPS, TLR4 signaling in immune cells activates transcription factor nuclear factor (NF)-κB, a process culminating in secretion of inflammatory cytokines such interleukin (IL)-1β, interferon (IFN)-γ, and tumor necrosis factor (TNF)-α [18,19,20].

The mitogen-activated protein kinase (MAPK) signaling pathway, which includes extracellular-signal-regulated kinase 1/2 (ERK)/MAPK, plays important role in immune cell differentiation and proliferation, along with expression of inflammatory cytokines [21]. Some innate immune responses also involve production of large amounts of reactive oxygen species (ROS) and nitric oxide (NO) [22]. NO is released at high concentrations by cytokine-activated macrophages [23]. Here, we used bone marrow-derived macrophages (BMDMs) and a murine macrophage cell line (RAW264.7) to examine the anti-inflammatory properties of SP. We also examined its effects on splenocytes, lymph nodes, and serum from LPS-injected C57BL/6 mice. 

## 2. Materials and Methods 

### 2.1. SP from Bombyx mori 

Acid-hydrolyzed SP was obtained from Worldway Co., Ltd. (Sejong, Korea). SP was derived from the cocoons of *Bombyx mori*. Silkworm cocoons were acid-hydrolyzed, neutralized, filtered, desalted, and freeze-dried. Finally, a yellow powder was prepared by spray drying. The nutrient composition of SP (100 g) is as follows: protein (86.8 g), carbohydrate (6.78 g), sodium (1.786 g), sugar (0.944 g), and other (3.69 g). The mean molecular weight measured by MicroQ-time-of-flight (TOF) III mass spectrometry (Bruker Daltonics, Hamburg, Germany) was 150–300. The free amino acid composition of SP was analyzed using an HPLC system comprising a pump (Waters 2695, Waters, Milford, MA, USA), an AccQ-Tag amino acid analysis column (Silica C18, 3.9 mm × 150 mm), and a Waters 2475 Multiλ Fluorescence detector. The free amino acid content was as follows: glycine (33.04%), alanine (28.09%), serine (11.09%), valine (2.67%), tyrosine (2.46%), aspartic acid (2.45%), glutamic acid (1.78%), threonine (1.22%), cysteine (1.04%), isoleucine (0.76%), proline (0.74%), leucine (0.72%), arginine (0.50%), phenylalanine (0.44%), lysine (0.38%), histidine (0.38%), and methionine (0.08%). SP powder was dissolved in distilled water for oral administration and cell treatment.

### 2.2. Reagents and Chemicals

The following materials were used in the study: LPS from *Escherichia coli* O111:B4, Griess reagent, and Giemsa solution (all from Sigma-Aldrich, St. Louis, MO, USA); antibodies specific for TLR4, p- mitogen-activated protein kinase 7 (TAK1), TAK1, p-IKK α/β, IKK α/β, IL-1β, COX-2, p-NF-κB, NF-κB, p-ERK1/2, and ERK1/2 (all from Cell Signaling Technology, Danvers, MA, USA); antibodies specific for iRAK4, TRAF6, SOD1, GPx1, p-Raf-1, and GAPDH (all from Santa Cruz Biotechnology, Dallas, TX, USA); IL-1β, TNF-α, IFN-γ cytokine kit, PerCP/Cyanine5.5 anti-mouse Ly6G, and Alexa-647 anti-mouse CD11b (all from BioLegend, San Diego, CA, USA); and PE anti-mouse Ly6C (BD Biosciences, Becton, NJ, USA).

### 2.3. Cell Culture and Treatment

Mouse RAW264.7 cells were obtained from the American Type Culture Collection (Manassas, VA, USA) and maintained at 37 °C/5% CO2 in Dulbecco’s modified Eagle’s medium supplemented with 10% fetal bovine serum and 1% penicillin/streptomycin at pH 7.2. Cells were washed and cultured in serum-free medium with or without SP and LPS.

### 2.4. BMDM Culture and Treatment

Bone marrow derived precursor cells were isolated from the experimental mouse as previously described [24]. The both femurs were separated, and the ends of the bones were cut off and the tissue was flushed by irrigation with PBS. The plugs were filtered by cell strainer (pore size 70 µm), and the cells recovered by centrifugation. The cells diluted in RPMI-1640 (containing L-glutamine and Na-bicarbonate) supplemented with 10% fetal bovine serum and 1% penicillin/streptomycin as growth medium and incubated at 37 °C and 5% CO_2_ in humidified incubator. The bone marrow derived precursor cells were allowed 7 d to adhere and differentiate into macrophages, and then the medium was refreshed. To perform the experiments in this paper, cells were cultured with or without SP and LPS.

### 2.5. Animals and Treatments

C57BL/6 mice (female, 4 weeks old) were purchased from Joongah Bio (Suwon, Korea) and housed at 20 ± 3 °C under a 12 h light-dark cycle. After 1 week of adaptation, mice were orally administrated SP (750 mg/kg/day) for 7 days. Optimal SP concentrations were selected based on body weight. Some mice received an intraperitoneal injection of LPS (2.5 mg/kg) and were euthanized 3 h later. Control mice were not injected with LPS.

### 2.6. Ethics Statement

All animals were cared for humanely according to the standards outlined in the “Guide for the Care and Use of Laboratory Animals” prepared by the National Academy of Sciences and published by the National Institutes of Health. All experiments were approved by the Institutional Animal Care and Use Committee (IACUC 180129) of CHA University (Seongnam, Kyunggi, Korea).

### 2.7. MTT Assay

Cells were seeded in 96-well plates at a density of 5 × 10^3^ per well and incubated overnight prior to treatment with SP (0, 25, 50, 100, or 200 µM) for an additional 24 h. MTT reagent was then added for 3 h. Supernatant was removed gently and 100 µL DMSO was added to extract the intracellular formazan. Cell viability was measured at 570 nm in a PowerWaveHT ELISA reader (BioTek, Winooski, VT, USA).

### 2.8. Flow Cytometry Analysis of Cell Populations

Spleens were obtained from C57BL/6 mice and homogenized. Single-cell suspensions were centrifuged for 10 min at 800× *g* and the supernatant was removed. Red blood cells were lysed in ACK lysis buffer (Lonza, Basel, Switzerland), centrifuged for 5 min at 800× *g*, and then washed twice with PBS. The cells were stained for 20 min with the following anti-mouse antibodies: Ly6G-PerCP/cy5.5, CD11b-Alexa647, and Ly6C-PE. Stained cells were washed twice with PBS and analyzed immediately by flow cytometry (CytoFlex; Beckman Coulter, Brea, CA, USA). Monocyte was determined by gating on viable forward vs side scatter gates. Compensation and data analysis were performed using FlowJo VX software (Ashland, OR, USA).

### 2.9. Western Blotting

Preparation of cell lysates, protein collection and concentration, electrophoresis, SDS-PAGE, and immunoblotting were performed as described preciously [25]. Primary and secondary antibodies were diluted 1000-fold and 500-fold, respectively. Signals were visualized using an EZ-Western Lumi Femto (DoGenBio, Seoul, Korea) and quantified using a LAS-4000 (GE Healthcare Life Sciences, Marlborough, MA, USA) [26]. Protein band intensities on each blot were quantified by densitometric analysis using ImageJ 1.48s software (Bethesda, MD, USA).

### 2.10. Griess Reagent Assay

Primary BMDMs and RAW264.7 cells were seeded (1 × 106 cells/well) and cultured overnight. Cells were pretreated for 3 h with SP (0, 50, or 100 µM). LPS (1 µg/mL) was added for another 18 h. The culture supernatants were collected and mixed with 100 µL Griess reagent (Merck Millipore, Burlington, MA, USA). After 10 min, the level of nitrite in the culture supernatant was measured by ELISA (570 nm).

### 2.11. ROS Assay

ROS were detected using 2′,7′-dichlorofluorescin diacetate (DCFDA; Cellular ROS Detection Assay Kit (Abcam, Cambridge, UK). Briefly, BMDMs and RAW264.7 cells were seeded (1 × 10^6^ cells/well) and cultured overnight. Cells were pretreated for 3 h with SP (0 or 100 µM), followed by LPS (1 µg/mL) for another 18 h. Cells were collected by trypsinization, washed with PBS, and centrifuged at 16,000× *g* for 10 min. Cells were suspended in culture medium containing 20 µM DCFDA and then incubated for 30 min at 37 °C. ROS concentration was measured by flow cytometry (CytoFlex; Beckman Coulter). Data were analyzed using FlowJo software (Ashland).

### 2.12. RNA Isolation from Lymph Nodes and Quantitative Real-Time PCR (qRT-PCR)

Lymph nodes were obtained from C57BL/6 mice and homogenized. Total RNA was extracted using Trizol reagent (Invitrogen, Carlsbad, CA, USA) and reverse transcribed to cDNA using the Maxime RT PreMix kit (Intron, Seongnam, Korea). The sequences of the oligonucleotide primers were as follows: TLR4, 5′-CAGAGTTGCTTTCAATGCCA-3′ (forward) and 5′-AGACTGTAATCAAGAAC CTG-3′ (reverse); iRAK4, 5′-AGCTGCGTCACCTACCTGTT-3′ (forward) and 5′-GTTTGGTGATGT TGCTGTGG-3′ (reverse); TRAF6, 5′-GATCGGGTTGTGTGTGTCTG-3′ (forward) and 5′-AGACACCCCAGCAGCTAAGA-3′ (reverse); IL-1β, 5′-CAGGATGAGGACATGAGCAC-3′ (forward) and 5′-CTCTGCACACTCAAACTCCA-3′ (reverse); pro-IL-1β, 5′-CTCACAAGCAGAG CACAAGC-3′ (forward) and 5′-AGAAGGATTTCATACCCGAC-3′ (reverse); IL-18, 5′-GGGTCAC AGCCAGTCCTCTT-3′ (forward) and 5′-TCAGACAACTTTGGCCGACT-3′ (reverse); iNOS, 5′-GGCTGTCAGAGCCTCGTGGC-3′ (forward) and 5′-CCCTTCCGAAGTTTCTGGCA-3′ (reverse); COX-2, 5′-CACTACATCCAGACCCACTT-3′ (forward) and 5′-ATGCTCCTGCTTGAGTATGT-3′ (reverse); and GAPDH, 5′-CAGAACTACATCCCTGCATC-3′ (forward) and 5′-CCACCTTCCTG ATGTCATCA-3′ (reverse). GAPDH was used as a control. QPCR was performed using the Mx3005P qPCR System (Agilent Technologies, Santa Clara, CA, USA).

### 2.13. Assessment of Cytokine Release

RAW264.7 and BMDM cells were pretreated for 3 h with SP (0, 50, or 100 µM), followed by LPS (1 µg/mL) for 18 h. The medium was collected to measure cytokine release. Mouse blood was collected by cardiac puncture at the time of euthanasia. IL-1β, TNF-α, and IFN-r levels in serum were investigated using appropriate cytokine kits.

### 2.14. Analysis of Peritoneal Lavage Fluid

Experimental mice were orally administrated SP (0 or 750 mg/kg/day) for 7 days and euthanized 3 h after the intraperitoneal injection of LPS (2.5 mg/kg). Peritoneal lavage fluid was isolated and centrifuged at 16,000× *g* for 15 min to separate cells from supernatant. Red blood cells were lysed in ACK lysis buffer (Lonza). The remaining cells were suspended in cold PBS, counted, and centrifuged onto microscope slides (800× *g* for 5 min) using a Thermo Cytospin 4 Cytocentrifuge (Thermo Fisher Scientific, Waltham, MA, USA). Slides were fixed in methanol, dried, and placed in diluted Giemsa solution for 30 min. Finally, stained slides were rinsed in deionized water and photographed under a Nikon Eclipse E600 microscope (Nikon Corporation, Tokyo, Japan).

### 2.15. Statistical Analysis

All data are presented as the mean ± the standard error of the mean (SEM). Statistical comparisons were evaluated by Student’s *t*-test or one-way analysis of variance (ANOVA) followed by Tukey’s or Duncan’s multiple range tests. Statistical analyses were performed using SPSS 20 (IBM, Armonk, NY, USA). A value of *p* < 0.05 was considered statistically significant.

## 3. Results

### 3.1. SP Exhibits Anti-Inflammatory Effects in LPS-Injected C57BL/6 Mice

To investigate the anti-inflammatory effects of SP in vivo, C57BL/6 mice received SP (0, 100, or 750 mg/kg/day) for 7 days, followed by an intraperitoneal (i.p.) injection of LPS (2.5 mg/kg) 3 h prior to euthanasia (Figure 1A). Spleen, lymph nodes, and serum were collected for analysis. First, we examined lymph nodes for expression of mRNA encoding key members of the TLR4 signaling pathway and inflammatory mediators (Figure 1B). After LPS injection, expression of mRNA encoding TRL4, iRAK4, TRAF6, IL-1β, pro-IL-1β, iNOS, and COX-2 was higher than that in the non-LPS-injected (control) group. However, oral administration of SP for 7 days reduced expression of mRNA encoding these genes to levels observed in the control group. We then examined protein expression in splenocytes; expression of TLR4, TRAF6, p-TAK1, IL-1β, and COX-2 mirrored that of the corresponding mRNA (Figure 1C). 

Pro-inflammatory cytokines play crucial roles during inflammatory responses [27,28,29]. To determine whether SP affects release of pro-inflammatory cytokines, mouse serum was collected by cardiac puncture at the time of the euthanasia and the amounts of IL-1β, TNF-α, and IFN-γ in serum were measured. Compared with the non-injected control group, LPS injection increased the concentrations of all of these cytokines. However, cytokine levels in the group orally administrated SP were similar to those in the control group (Figure 1D). Thus, SP has a marked anti-inflammatory effect in LPS-stimulated C57BL/6 mice.

### 3.2. SP Reverses the LPS-Mediated Increases in the Monocyte and Macrophage Populations in LPS-Injected C57BL/6 Mice

Next, we used flow cytometry to examine the effects of SP on monocyte populations in the spleen; these cells differentiate into macrophages in response to immune stimulation [30]. The number of Ly6G-CD11b+Ly6C^hi^ monocytes in LPS-injected mice was higher than that in the control group. However, oral administration of SP reduced these numbers significantly (Figure 2A). 

Furthermore, we performed peritoneal lavage with PBS and collected the washed-out cells. We then performed cytospin counts on Giemsa-stained slides to ascertain the percentage of macrophages and the total number of peritoneal lavage cells. The macrophage population in the peritoneum of LPS-injected mice was much higher than that in non-injected mice. By contrast, oral administration of SP for 7 days led to a marked reduction in the number of peritoneal macrophages (Figure 2B). These results suggest that SP inhibits the LPS-induced increase in the number of peritoneal monocytes, leading to a reduction in the macrophage population.

### 3.3. Anti-Inflammatory Effects of SP on BMDMs Are Mediated Via the TLR4 Signaling Pathway

Since SP affects macrophage numbers in LPS-injected mice, we next investigated the anti-inflammatory effects of the SP on BMDMs obtained from the C57BL/6 mice. Primary cultured BMDMs were pretreated with SP for 3 h and then incubated with LPS for another 18 h (Figure 3A,B). LPS increased expression of TLR4 and TRAF6 when compared with that in control cells. By contrast, SP reduced expression of these proteins in a concentration-dependent manner (Figure 3A). Moreover, stimulation with LPS increased phosphorylation of TAK1 and IKK α/β; phosphorylation was suppressed by high concentrations of SP without affecting the amount of total TAK1 and IKK α/β (Figure 3B). Phosphorylation of NF-κB triggers translocation to the nucleus, where it acts as a transcriptional factor for inflammatory genes [31,32]; therefore, we measured phosphorylation of NF-κB every 30 min. Phosphorylation of NF-κB peaked 30 min after LPS treatment; however, this was inhibited markedly by SP pretreatment without affecting the amount of non-phosphorylated NF-κB (Figure 3C). These data show that SP down-regulates expression of key factors in the TLR4 signaling pathway that are upregulated by LPS, implying that SP exerts its anti-inflammatory effects by modulating the TLR4 signaling pathway.

### 3.4. SP Prevents LPS-Induced Generation of Pro-Inflammatory Cytokines and Oxidative Stress

When activated NF-κB translocates to the nucleus, it binds to the promoter sites of target genes and induces transcription of pro-inflammatory cytokines and mediators, including IL-1β, iNOS, and COX-2 [33]. Therefore, we investigated the inhibitory effects of SP on these pro-inflammatory cytokines and mediators by western blotting. IL-1β, iNOS, and COX-2 protein were undetectable in unstimulated BMDMs. However, they were noticeably expressed after exposure to LPS; expression of these molecules was suppressed by SP in a concentration-dependent manner (Figure 4A).

To determine whether SP affects production of cytokines, we collected culture supernatant from BMDMs. After incubation with LPS alone for 18 h, the levels of IL-1β, IFN-γ, and TNF-α were significantly higher than those in the control group; by contrast, co-incubation with SP and LPS for 18 h reduced expression of these cytokines (Figure 4B). LPS triggers inflammatory responses via release of superoxide, ROS, and NO; therefore, we investigated whether SP affects oxidative stress. As expected, LPS treatment inhibited expression of SOD1 and GPx1, which protect against oxidative stress; however, SP upregulated expression of these molecules (Figure 4C). ROS produced by BMDMs pretreated with SP for 3 h and treated with LPS for another 18 h were measured by flow cytometry. LPS increased the DCFH-DA signal intensity, which was attenuated by SP pretreatment (Figure 4D). Moreover, we assessed NO levels in culture medium from BMDMs. The Griess reagent assay showed that SP prevented the LPS-induced increase in NO release in a concentration-dependent manner (Figure 4E). Taken together, the data suggest that SP inhibits NF-κB-mediated increases in pro-inflammatory molecules and exhibits antioxidant effects.

### 3.5. SP Inhibits LPS-Stimulated Transduction of TLR4 Signals and Phosphorylation of MAPKs in RAW264.7 Macrophages

To investigate the cytotoxic effects of SP in RAW264.7 cells, we exposed cells to various concentrations of SP (0, 25, 50, 100, or 200 μM) for 24 h. As shown in Figure 5A, SP was not cytotoxic up to 100 μM. Therefore, we selected SP concentrations of 0, 50, and 100 μM for subsequent in vitro experiments. To obtain a mechanistic understanding of the way in which SP-mediated inflammatory responses are regulated, we investigated expression of proteins downstream of TLR4 in RAW264.7 cells. Cells were pretreated with SP for 3 h and incubated with LPS for another 18 h (Figure 5B,C). LPS-stimulated expression of TLR4, iRAK4, and TRAF6 in RAW264.7 cells was inhibited significantly by SP (Figure 5B). Forced phosphorylation of TAK1 and IKK α/β by LPS also fell in the presence of SP without affecting the total amounts of these molecules (Figure 5C). Next, we assessed expression of NF-κB at intervals of 30 min to investigate how SP affects activation and translocation of NF-κB (Figure 5D). Expression of p-NF-κB, which peaked 30 min after LPS stimulation, was reduced by SP without affecting the total amount of NF-κB. TLR4 signaling activates the MAPKs family via phosphorylation of Raf-1 and ERK [34,35,36]. Phosphorylation of Raf-1 in LPS-treated cells was greater than that in control cells, but phosphorylation of Raf-1 was markedly inhibited in cells pretreated with SP (Figure 5E). Because phosphorylated ERK1/2 translocates to the nucleus, we measured expression of p-ERK1/2 at 30 min intervals. Expression of p-ERK1/2 was highest at 30 min after LPS stimulation; however, SP inhibited phosphorylation of ERK1/2 at 60 min post-LPS treatment (Figure 5F). Thus, SP suppresses LPS-induced inflammation via the TLR4 signaling pathway and MAPK signaling pathways in RAW264.7 cells.

### 3.6. SP Suppresses Release of Pro-Inflammatory Cytokines and Inflammatory Responses by RAW264.7 Macrophages

To further examine the anti-inflammatory effects of SP on LPS-treated RAW264.7 cells, we pretreated cells with SP and then stimulated them with LPS for 18 h. LPS-stimulated expression of IL-1β, iNOS, and COX-2 proteins, which are considered typical pro-inflammatory cytokines and inflammatory mediators, was reduced markedly by SP (Figure 6A). Also, SP pretreatment led to a significant decrease in the level of IL-1β, IFN-γ, and TNF-α in the culture medium of LPS-stimulated RAW264.7 cells (Figure 6B). To investigate production of NO in RAW264.7 cells, cells were pretreated for 3 h with increasing concentrations of SP and then treated with LPS for another 18 h. The Griess reagent assay revealed that increasing concentrations of SP inhibited the LPS-induced increase in NO production (Figure 6C). LPS decreased expression of SOD1 and GPx1; however, this effect was reversed by SP (Figure 6D). Finally, to determine whether SP has antioxidant effects on LPS-stimulated RAW264.7 cells, we performed flow cytometry analysis of cells treated with DCFH-DA. DCFH-DA signals in RAW264.7 cells increased after LPS treatment but were attenuated significantly by SP (Figure 6E). These results indicate that SP suppresses LPS-induced release of pro-inflammatory cytokines and LPS-induced oxidative stress. Consistent with the previous results, these data show that SP exerts anti-inflammatory effects through the TLR4 signaling pathway.

## 4. Discussion

Here, we asked whether SP modulated TLR4 signaling and subsequent inflammatory responses by immune cells in vivo and in vitro. First, we examined the effect of SP in an LPS-treated C57BL/6 mouse model. Oral administration of SP reversed LPS-induced inflammatory responses in the spleen, lymph nodes, and blood (Figure 1A). Mechanistic analysis of LPS-stimulated primary BMDMs from C57BL/6 mice and of RAW264.7 cells revealed that SP exerted its anti-inflammatory effects through the TLR4 signaling pathway. 

Monocytes are important effectors and regulators of inflammation triggered by LPS [37,38]. Monocytes are rapidly recruited to the region where the inflammatory response occurs, and they differentiate into inflammatory subsets such as macrophages [39]. For flow cytometry analysis, Ly6G-cells were gated to exclude neutrophils, and CD11b+ and Ly6Chi cells were gated to include myeloid cell and monocytes, respectively [40,41]. We found that SP down-regulated the LPS-induced increase in the number of Ly6G-CD11b+Ly6C^hi^ monocytes in the splenocyte population (Figure 2A). Inflammatory Ly6C^hi^ monocytes circulating in the bone marrow and spleen give rise to macrophages [39,42,43], which are representative innate immune cells that play a central role in immune regulation and host defense [44,45]. The peritoneal lavage assay revealed that SP reversed the LPS-induced increase in peritoneal macrophage numbers (Figure 2B). For further mechanistic analysis, we used BMDM from C57BL/6 mice and RAW264.7 macrophage cells in subsequent experiments.

The TLR family, especially TLR4, is a class of proteins that plays a key role in the innate immune system [46]. TLR4 is a PRR found in *Drosophila melanogaster* and mammals; this PRR binds to the PAMP LPS, which was expressed and shed by Gram-negative bacteria [9,47,48]. We examined the spleen, lymph nodes, and serum from SP-treated mice to determine whether SP suppresses LPS-induced inflammation through TLR4 signaling. SP decreased LPS-mediated expression of TLR4, iRAK4, TRAF6, and TAK1 at the mRNA and protein levels. Significant differences were noted between control and test animals at both low (100 mg/kg/day) and high (750 mg/kg/day) SP concentrations (Figure 1B,C). Since the spleen and lymph nodes are innate immune organs that harbor active macrophages, we used BMDMs and RAW264.7 cells to examine the effect of SP on TLR4 signaling. We found that SP inhibits LPS-induced inflammation by downregulating expression of TLR4 signaling molecules TLR4, TRAF6, TAK1, and IKKα/β in both BMDM and RAW264.7 cells (Figure 3A,B and Figure 5B,C). TLR4 bind to LPS to induce activation and translocation of NF-κB to the cell nucleus [49]. Therefore, we examined LPS-induced expression of NF-κB at 0, 30, and 60 min since translocation occurs within 1 h after the LPS stimulus [49,50]. SP reduced LPS-induced expression of NF-κB in BMDMs and RAW264.7 cells in 30 min (Figure 3C and Figure 5D). TLR4 signaling activates MAPK signaling, which also plays key role in regulating inflammation [51]. We also found that SP reduced LPS-induced expression of Raf-1, a member of the ERK pathway (Figure 5E). SP suppressed LPS-induced phosphorylation of ERK at 30 and 60 min (Figure 5F). Totally, SP also indicated the possibility of regulating inflammatory response by LPS through MAPK signaling suppression activated by LPS-indicated TLR4 signaling.

Finally, since NF-κB induces expression of pro-inflammatory genes, including those encoding cytokines and mediators, we investigated whether SP regulates expression of inflammatory cytokines [52]. SP decreased LPS-induced expression of genes encoding pro-inflammatory mediators IL-18, IL-1β, iNOS, and COX-2 in lymph nodes from C57BL/6 mice. The same was also true for the products of these genes. This was confirmed in vitro using BMDMs and RAW264.7 cells (Figure 1C, Figure 4A and Figure 6A). In addition, SP reduced expression of LPS-induced inflammatory cytokines IL-1β, IFN-γ, and TNF-α in the serum of C57BL/6 mice (Figure 1D) and in the supernatant of BMDM and RAW264.7 cells (Figure 4B and Figure 6B). In addition, LPS is known to induce oxidative stress through a significant decrease in antioxidant enzymes along with an increase in ROS and NO levels [53]. An imbalance between the ROS generation and their elimination by protective mechanisms causes inflammation [54]. Expression of genes encoding SOD1 and GPx1, which protect against oxidative stress, in BMDMs and RAW 264.7 cells increased after SP treatment (Figure 4C and Figure 6D). SP also significantly inhibited LPS-induced ROS and NO production (Figure 4D,E and Figure 6C,E). Therefore, SP suppressed LPS-induced inflammation by improving oxidative imbalance.

This paper demonstrates the efficacy of anti-inflammatory response through in vitro using RAW 264.7 cells, BMDMs isolated from LPS-treated mice, and in vivo using the spleen, and lymph nodes of injected mice and by measuring cytokine concentrations in mouse serum. Taken together, the above data suggest that SP has potential as a novel food supplement for treatment of acute inflammatory diseases.

## Figures and Tables

**Figure 1 biomolecules-10-00771-f001:**
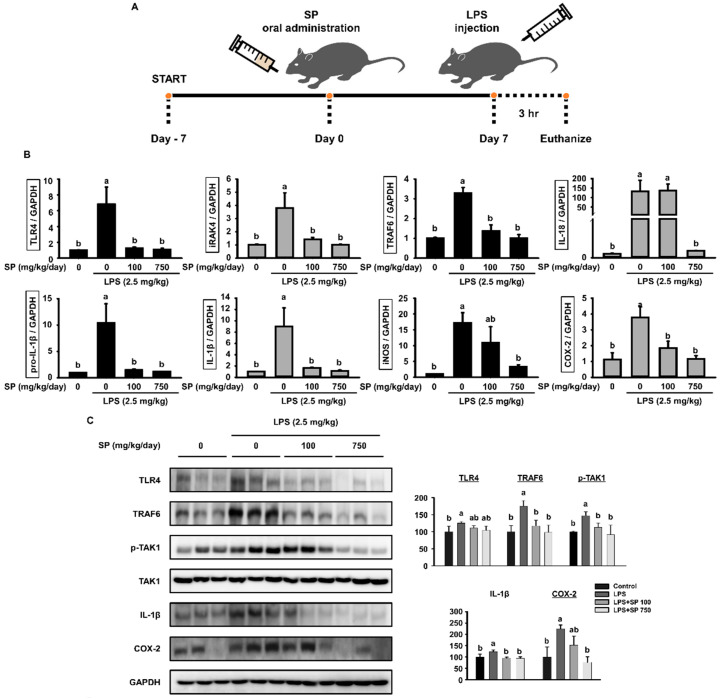
Silk peptide (SP) has anti-inflammatory effects in lipopolysaccharide (LPS)-injected C57BL/6 mice. (**A**) Experimental protocol: After 1 week of adaptation, C57BL/6 mice were orally administered SP (750 mg/kg/day) for 7 days. On Day 7, mice were injected LPS (2.5 mg/kg) for 3 h prior to euthanize. (**B**) RNA was isolated from lymph nodes and quantitative reverse transcription polymerase chain reaction (RT-PCR) analysis was performed (mean ± SEM; *n* = 3). (**C**) Protein was isolated from the spleen and western blot analysis was performed (mean ± SEM; *n* = 3). GAPDH served as a control. Data were quantitatively analyzed using ImageJ software. (**D**) Serum was collected from experimental C57BL/6 mice at the time of euthanasia, and levels of IL-1β, interferon (IFN)-γ, and tumor necrosis factor (TNF)-α were measured by ELISA (mean ± SEM; *n* = 4). Statistical significance was determined using one-way ANOVA followed by Duncan’s test. Datasets denoted by different letters are significantly different (*p* < 0.05, a > ab > b > c).

**Figure 2 biomolecules-10-00771-f002:**
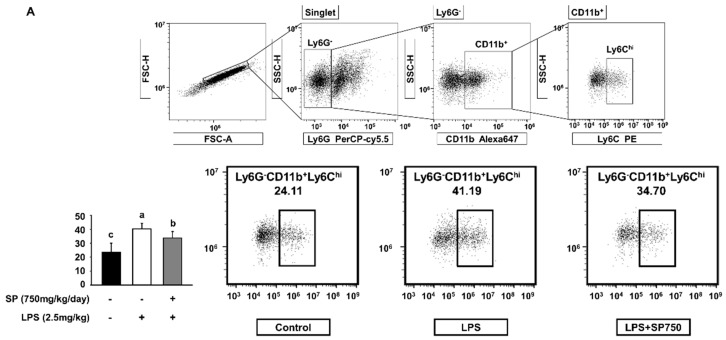
SP reduces the population of monocytes and macrophages in the peritoneum of LPS-injected C57BL/6 mice. (**A**) Splenic cells from experimental C57BL/6 mice were stained with anti-Ly6G-PerCP/Cy5.5, anti-CD11b-Alexa647, and anti-Ly6C-PE. Ly6G-cells were gated to exclude neutrophils. The gate was then placed on CD11b^+^ cells (myeloid cells). Inflammatory monocyte compartments were identified as Ly6G^−^CD11b^+^Ly6C^hi^. The numbers within the plots represent the percentage of cell population (mean ± SEM; *n* = 4). The gating based on the reactivity of antibody relative to each isotype control. The fluorescence scale is logarithmic. (**B**) Representative images of cytospin slides of peritoneal lavage cells collected from experimental C57BL/6 mice. The white boxes denote macrophages. The number of macrophages and total cell numbers were counted (mean ± SEM; *n* = 3). Statistical significance was determined using one-way ANOVA followed by Tukey’s post-hoc test. Datasets denoted by different letters are significantly different (*p* < 0.05, a > b > c).

**Figure 3 biomolecules-10-00771-f003:**
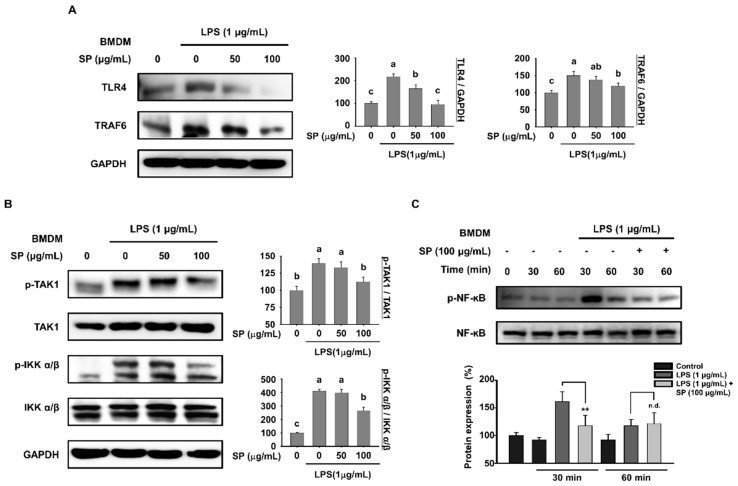
Anti-inflammatory effects of SP in bone marrow derived macrophages (BMDMs) are mediated via the toll like receptor 4 (TLR4) signaling pathway. BMDMs pretreated with SP (0, 50, and 100 µg/mL) for 3 h and stimulated with LPS (1 µg/mL) for another 18 h were subjected to western blot analysis to detect (**A**) TLR4 and TRAF6, and (**B**) p-TAK1, TAK1, p-IKK α/β, IKK α/β, and GAPDH (mean ± SEM; *n* = 3). TAK1 and IKK α/β were used as controls for p-TAK1 and p-IKK α/β, respectively. Data were analyzed quantitatively using ImageJ software. (**C**) Expression of p-NF-κB and total NF-κB in BMDMs pretreated with SP (100 µg/mL) for 3 h and stimulated with LPS (1 µg/mL) for the indicated times was examined by western blotting. NF-κB was used as a control for p-NF-κB (mean ± SEM; *n* = 3, ***p* < 0.01). Statistical significance was determined using Student’s t-test or one-way ANOVA followed by Duncan’s test. Datasets denoted by different letters are significantly different (*p* < 0.05, a > b > c).

**Figure 4 biomolecules-10-00771-f004:**
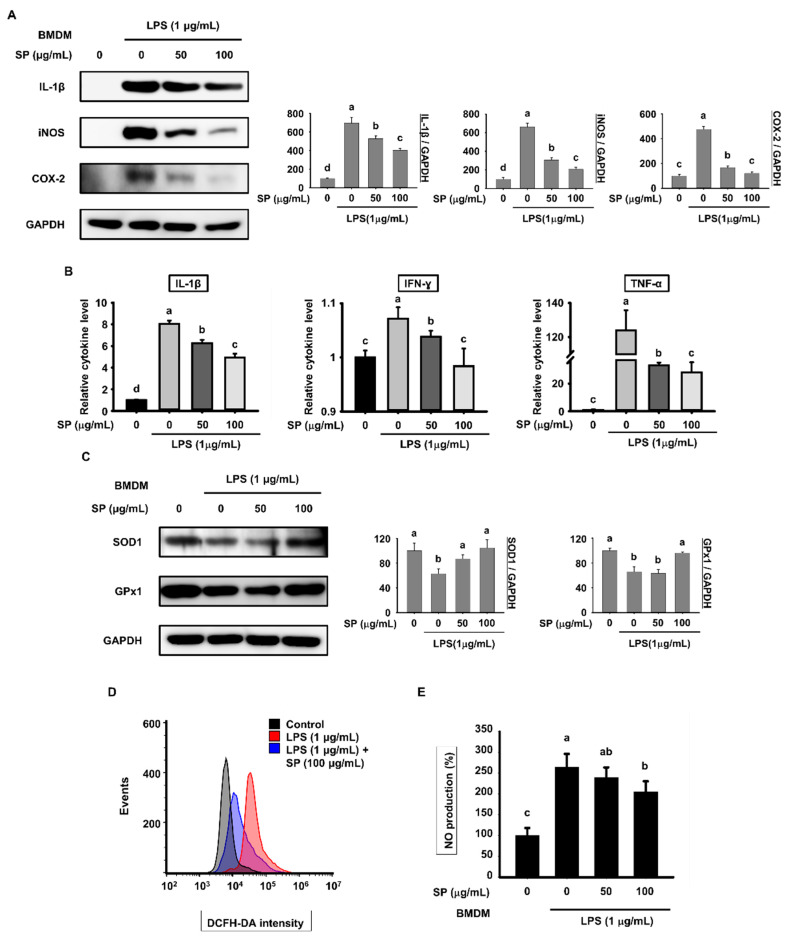
Effects of SP on pro-inflammatory cytokines and oxidative stress in BMDMs. (**A**) BMDMs pretreated with SP (0, 50, and 100 µg/mL) for 3 h and stimulated with LPS (1 µg/mL) for another 18 h were subjected to western blot analysis to detect pro-inflammatory cytokine and mediators (mean ± SEM; *n* = 3). GAPDH served as a control. Data were analyzed quantitatively using ImageJ software. (**B**) The culture medium from BMDMs was collected and the levels of IL-1β, IFN-γ, and TNF-α were measured by ELISA (mean ± SEM; *n* = 4). (**C**) Expression of SOD1 and GPx1 in BMDMs pretreated with SP (0, 50, and 100 µg/mL) for 3 h and then stimulated with LPS (1 µg/mL) for 18 h was examined by western blotting. (**D**) ROS production by BMDMs pretreated with SP (100 µg/mL) for 3 h and treated with LPS (1 µg/mL) for 18 h was measured by flow cytometry following incubation with DCFH-DA (mean ± SEM; *n* = 3). (**E**) BMDMs were pretreated with SP (0, 50, and 100 µg/mL) for 3 h and then stimulated with LPS (1 µg/mL) for 18 h, and the level of NO in the culture medium was assessed in a Griess reagent assay (mean ± SEM; *n* = 3). Statistical significance was determined using a one-way ANOVA followed by Duncan’s test. Datasets denoted by different letters are significantly different (*p* < 0.05, a > b > c > d).

**Figure 5 biomolecules-10-00771-f005:**
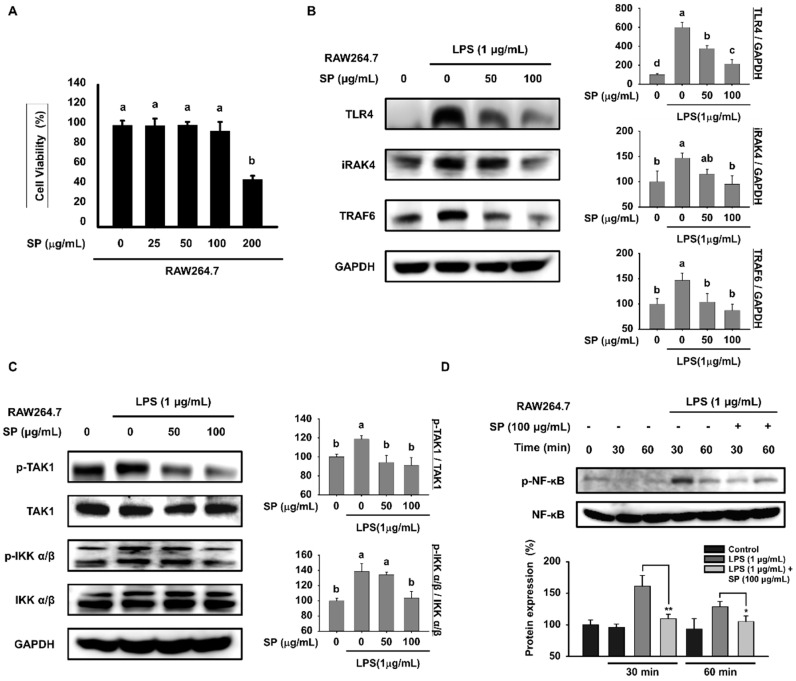
SP inhibits LPS-stimulated TLR4 signal transduction and phosphorylation of MAPKs in RAW264.7 cells. (**A**) RAW264.7 macrophages were treated for 24 h with 0, 25, 50, 100, or 200 μM SP in medium containing 1% serum. Cell viability was measured in an MTT assay after 24 h (mean ± SEM; *n* = 5). (**B**) RAW264.7 cells were pretreated with SP (0, 50, and 100 µg/mL) for 3 h and then stimulated with LPS (1 µg/mL) for another 18 h prior to western blot analysis of (**A**) TLR4, iRAK4, and TRAF6, and (**C**) p-TAK1, TAK1, p-IKK α/β, IKK α/β, and GAPDH (mean ± SEM; *n* = 3). TAK1 and IKK α/β were used as controls for p-TAK1 and p-IKK α/β, respectively. Data were analyzed quantitatively using ImageJ software. (**D**) Expression of p-NF-κB and NF-κB in RAW264.7 cells pretreated with SP (100 µg/mL) for 3 h and stimulated with LPS (1 µg/mL) for the indicated times was measured by western blotting. NF-κB was used as a control for p-NF-κB (mean ± SEM; *n* = 3, **p* < 0.05, ***p* < 0.01). (**E**) RAW264.7 cells were pretreated with SP (0, 50, and 100 µg/mL) for 3 h and then stimulated with LPS (1 µg/mL) for another 18 h prior to western blot analysis of p-Raf-1. GAPDH was used as a control (mean ± SEM; *n* = 3). (**F**) Expression of p-ERK1/2 and ERK1/2 in RAW264.7 cells pretreated with SP (100 µg/mL) for 3 h and then stimulated with LPS (1 µg/mL) for the indicated times was assessed by western blotting. NF-κB was used as a control for p-NF-κB (mean ± SEM; *n* = 3, **p* < 0.05, ****p* < 0.005). Statistical significance was determined using the Student’s *t*-test or one-way ANOVA followed by Duncan’s test. Datasets denoted by different letters are significantly different (*p* < 0.05, a > b > c).

**Figure 6 biomolecules-10-00771-f006:**
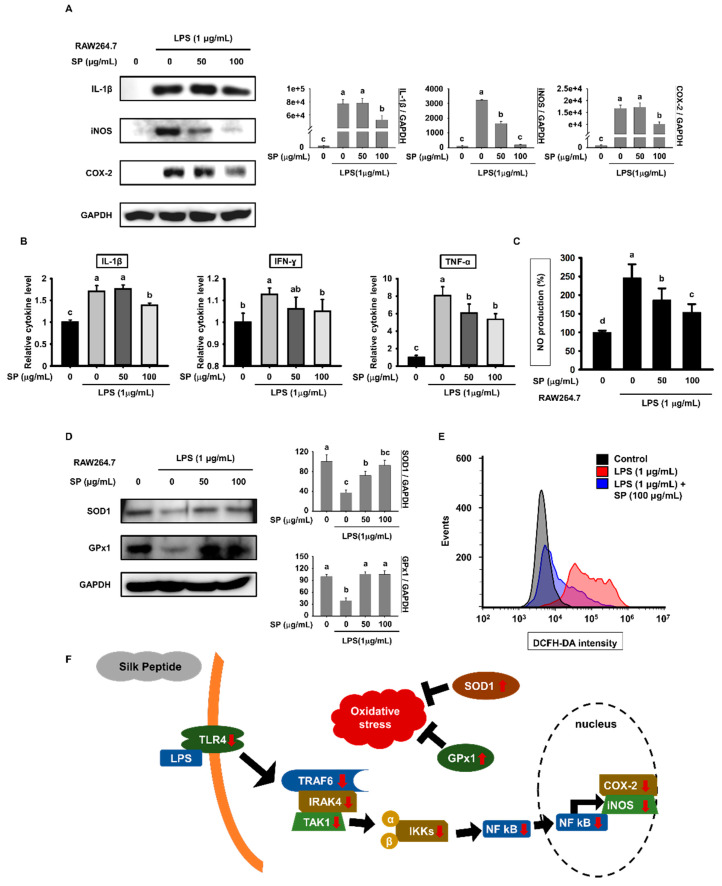
Effects of SP on production of pro-inflammatory cytokines and inflammatory responses by RAW264.7 macrophages. RAW264.7 cells were pretreated with SP (0, 50, and 100 µg/mL) for 3 h and then treated with LPS (1 µg/mL) for another 18 h. (**A**) Expression of IL-1β, iNOS, and COX-2 by RAW264.7 cells was determined by western blotting (mean ± SEM; *n* = 3). GAPDH was used as a control. (**B**) Cytokine levels in the culture medium of RAW264.7 cells were estimated (mean ± SEM; *n* = 3). (**C**) Levels of NO in the culture medium of RAW264.7 cells were analyzed using Griess reagent (mean ± SEM; *n* = 3). (**D**) Expression of SOD1 and GPx1 in RAW264.7 cells was measured by western blot analysis (mean ± SEM; *n* = 3). GAPDH was used as a control. (**E**) Levels of ROS in RAW264.7 cells pretreated with SP (100 µg/mL) for 3 h and then treated with LPS (1 µg/mL) for 18 h were measured by flow cytometry following incubation with DCFH-DA (mean ± SEM; *n* = 3). (**F**) Graphical abstract. Statistical significance was determined using one-way ANOVA followed by Duncan’s test. Datasets denoted by different letters are significantly different (*p* < 0.05, a > b > c > d).

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
