# Peer review of "Dietary Silk Peptide Inhibits LPS-Induced Inflammatory Responses by Modulating Toll-Like Receptor 4 (TLR4) Signaling"

_biomolecules, 2020, doi:10.3390/biom10050771_

Round 1

Reviewer 1 Report

The wok of Chei et al is the first report that SP inhibits inflammatory responses by modulating TLR4 signaling. This work opens avenues to understand whether SP is a TLR4 antagonist. The data are solid and the manuscript is well written.

I miss in this work:

1- mouse TLR4 and human TLR4 have different activators/modulators. Considering the applications of SP to humans, it will be interesting that authors show that SP inhibits the activation of human macrophages.

2- The discussion just summarizes the result section rather than discuss the conclusions of the work with other authors. Few references are there to support general knowledge. The authors should improve the discussion further. 

3- The reader will benefit of an illustrative picture shown the complete TLR4 signaling to understand which intracellular molecules were tested and which others were not.

4-  Why authors explored TLR4 activation mediated proinflammatory pathway rather than other TLRs?, did not they find regulation of other TLRs?. If so, I think these data are very valuable.

Technical issues

1-GADPH is used as a house-keeping gene (invariable control) in RT-PCR assays. In Figure 1C GAPDH production is increased with the concentration of SP. It seems that GAPDH is not a useful control.

2- What does it mean ab in Fig 1B?, this is not described in the legend

3- There is a large variation in gene expression of genes encoding for TLR4, IRAK, IL-18, pro-IL-1B, and IL1B data in Fig 1B. Even in some cases, such variation is up to 50% of the mean. Note that authors show SEMs rather standard variation. I am surprised that comparisons between groups are significant. Even more confusing is that protein production did not reflect such variation (Fig. 1C).

4- Which cell controls were used to discriminate between the different cell populations in Fig 2A?, It seems that authors discriminated based on the reactivity of antisera but they did not use unspecific antisera or a control population for gating.

5-Authors explored the activation of MAPSK, ERK, Raf-1 and NF-kB which became activated via TLR4 through the TIRAP dependent cascade, but what about TRAM-dependent cascade?. Did the authors test for instance IRF3 production?

In line 237 authors conclude  ¨SP reverses the LPS induced increase..¨, however, SP was administrated before LPS. Please rephrase the conclusion.

Author Response

Reviewer1

The wok of Chei et al is the first report that SP inhibits inflammatory responses by modulating TLR4 signaling. This work opens avenues to understand whether SP is a TLR4 antagonist. The data are solid and the manuscript is well written.

I miss in this work:

1- mouse TLR4 and human TLR4 have different activators/modulators. Considering the applications of SP to humans, it will be interesting that authors show that SP inhibits the activation of human macrophages.

  • Thank you for your comments. Unfortunately, there was no human macrophage cell lines in our lab, so we couldn’t test the effectiveness of SP on human macrophages. However, we determined not only in vivo study, but also, BMDM and mouse macrophage cell line to confirmed our research.

2- The discussion just summarizes the result section rather than discuss the conclusions of the work with other authors. Few references are there to support general knowledge. The authors should improve the discussion further.

  • Thank for your suggestion. We improved the discussion (reference 40, 52, and 54).

3- The reader will benefit of an illustrative picture shown the complete TLR4 signaling to understand which intracellular molecules were tested and which others were not.

  • According to your kind comment, we added graphical abstract to better understanding in Figure 6F.

4-  Why authors explored TLR4 activation mediated proinflammatory pathway rather than other TLRs?, did not they find regulation of other TLRs?. If so, I think these data are very valuable.

  • To study the effect of SP on the innate immune system, we used LPS, which is the most commonly used as stimulant in the innate immune analysis. As documented in introduction, LPS is a PAMP found in the outer membrane of gram-negative bacteria and it binds to Toll-like receptor 4(TLR4) expressed on the surface of immune cells. So, we focused on TLR4 activation induced by LPS rather than other TLRs.

Technical issues

  • GADPH is used as a house-keeping gene (invariable control) in RT-PCR assays. In Figure 1C GAPDH production is increased with the concentration of SP. It seems that GAPDH is not a useful control.
  • Your comment is right, but we showed quantification graphs that represent the expression of each protein RELATIVE to from GAPDH.
  • What does it mean ab in Fig 1B?, this is not described in the legend
  • As your comment, we defined ‘ab’ in line 223.
  • There is a large variation in gene expression of genes encoding for TLR4, IRAK, IL-18, pro-IL-1B, and IL1B data in Fig 1B. Even in some cases, such variation is up to 50% of the mean. Note that authors show SEMs rather standard variation. I am surprised that comparisons between groups are significant. Even more confusing is that protein production did not reflect such variation (Fig. 1C).
  • Thanks for your constructive comments. We revised Figure 1B to SEMs.
  • Which cell controls were used to discriminate between the different cell populations in Fig 2A?, It seems that authors discriminated based on the reactivity of antisera but they did not use unspecific antisera or a control population for gating.
  • We used flow cytometry to examine the effects of SP on monocyte populations in the mouse spleen. As shown in above of the Figure 2A, single cells were gated, Ly6G- cells were gated to exclude neutrophils, and CD11b+ and Ly6Chi cells were gated to include myeloid cell and monocytes, respectively. The three groups of graphs representing monocyte(Ly6G- CD11b+ Ly6Chi), below Figure 2a, were used same gating strategy, as described earlier. Of course, every gating was based on the reactivity of antibody relative to each isotype control. We also added the additional explanation, “The gating based on the reactivity of antibody relative to each isotype control.” (line 246).
  • Authors explored the activation of MAPSK, ERK, Raf-1 and NF-kB which became activated via TLR4 through the TIRAP dependent cascade, but what about TRAM-dependent cascade?. Did the authors test for instance IRF3 production?
  • As your comments, TLR4 activates both TIRAP-dependent cascade and TRAM-dependent cascade. We explored the activation of the molecules included in these two cascades, but the molecules in TRAM-dependent cascade, such as TRAM and IRF3, showed no changes by SP treatment. So, we focused on TLR4 signaling pathway via TIRAP-dependent cascade.

In line 237 authors conclude ¨SP reverses the LPS induced increase..¨, however, SP was administrated before LPS. Please rephrase the conclusion

  • According to your comment, we edited the sentence in line 236-237

Reviewer 2 Report

Very well written and interesting manuscript.  The hypothesis is sound and the experimental design is sound.  I found just a few minor issues in the Introduction and M&M that could be changed by the authors:

1.  Line 29: the citation for [3] is repeated 7 times

2.  Line 45: not all TLR are membrane receptors, some are intracellular endosomal.  Be careful about making blanket statements

3.  Line 56-57: same here: ROS and NO are produced by macrophages, but the narrative suggests ALL innate immune cells produce both and that is not true.  NK cells produce neither, neutrophils are no major producers of NO.

4.  In vivo LPS concentration:  Feeding 750mg/kg/day is huge.  One can kill a mouse with microgram amounts.  How did the authors come up with this dose because, even for only three hours, this dose borders on unethical and painful to the animals.  Similarly ip injection of 2.5 mg of LPS?  The literature is full of papers describing the toxic effects of microgram amounts.  Please check your dosages.

5.  I understand the decrease in numbers of cells migrating to the peritoneal cavity by the treatment since the SP in neutralizing the inflammatory effects, but was there really a decrease on the number of total cells in the circulation of just a decrease in activated cells (based on markers)?  did the SP treatment actually reduce the number of cells?

Author Response

Reviewer2

Very well written and interesting manuscript.  The hypothesis is sound and the experimental design is sound.  I found just a few minor issues in the Introduction and M&M that could be changed by the authors:

  1. Line 29: the citation for [3] is repeated 7 times
  • We revised it (line 29).
  1. Line 45: not all TLR are membrane receptors, some are intracellular endosomal. Be careful about making blanket statements
  • Thanks for your kind comment. We edited the sentence in line 46-47.
  1. Line 56-57: same here: ROS and NO are produced by macrophages, but the narrative suggests ALL innate immune cells produce both and that is not true. NK cells produce neither, neutrophils are no major producers of NO.
  • According your comment, we revised the sentence in line 56.
  1. In vivo LPS concentration: Feeding 750mg/kg/day is huge.  One can kill a mouse with microgram amounts.  How did the authors come up with this dose because, even for only three hours, this dose borders on unethical and painful to the animals. Similarly ip injection of 2.5 mg of LPS?  The literature is full of papers describing the toxic effects of microgram amounts.  Please check your dosages.
  • We appreciate your constructive comments. As we mentioned in 2.5 Animals and treatments, mice were orally administrated (750 mg/kg/day) for 7 days and received an intraperitoneal injection of LPS (2.5 mg/kg) for 3 h. We selected optimal SP concentrations based on mice body weight, moreover, referred to the paper, Jang et al., the SP concentration of 750 mg/kg/day is not toxic for mice [1]. Also, the concentration and time of LPS injection was set as the optimum condition for our study by referring to several papers [2-5].
  1. I understand the decrease in numbers of cells migrating to the peritoneal cavity by the treatment since the SP in neutralizing the inflammatory effects, but was there really a decrease on the number of total cells in the circulation of just a decrease in activated cells (based on markers)? did the SP treatment actually reduce the number of cells?
  • In Figure 1B, we counted both the number of macrophages (based on morphology) and total number of cells, and the right graph of Figure 1B showed the number of macrophages relative to the total number of cells. Indeed, SP treatment did not reduce the total number of cells (total number of cells are similar in all groups), but the number of macrophages increased by LPS and decreased by SP treatment. These results suggest that SP reverses the LPS-induced increase in the number of peritoneal macrophages, without affecting the total number of cells. Here is our raw data (data not shown in manuscript).

Control

LPS

LPS+SP750

# of total cell

47

58

56

# of macrophage

4

17

8

# of macrophage /# of total cell

0.09

0.29

0.14

# of macrophage /# of total cell *100

8.51

29.31

14.29

References

  1. Jang, S.H.; Oh, M.S.; Baek, H.I.; Ha, K.C.; Lee, J.Y.; Jang, Y.S. Oral Administration of Silk Peptide Enhances the Maturation and Cytolytic Activity of Natural Killer Cells. Immune Netw 2018, 18, e37, doi:10.4110/in.2018.18.e37.
  2. Starr, M.E.; Ueda, J.; Takahashi, H.; Weiler, H.; Esmon, C.T.; Evers, B.M.; Saito, H. Age-dependent vulnerability to endotoxemia is associated with reduction of anticoagulant factors activated protein C and thrombomodulin. Blood 2010, 115, 4886-4893, doi:10.1182/blood-2009-10-246678.
  3. Cho, M.J.; Kim, J.H.; Park, C.H.; Lee, A.Y.; Shin, Y.S.; Lee, J.H.; Park, C.G.; Cho, E.J. Comparison of the effect of three licorice varieties on cognitive improvement via an amelioration of neuroinflammation in lipopolysaccharide-induced mice. Nutr Res Pract 2018, 12, 191-198, doi:10.4162/nrp.2018.12.3.191.
  4. Hsu, D.Z.; Chu, P.Y.; Liu, M.Y. The non-peptide chemical 3,4-methylenedioxyphenol blocked lipopolysaccharide (LPS) from binding to LPS-binding protein and inhibited pro-inflammatory cytokines. Innate Immun 2009, 15, 380-385, doi:10.1177/1753425909341806.
  5. Salminen, A.; Paananen, R.; Vuolteenaho, R.; Metsola, J.; Ojaniemi, M.; Autio-Harmainen, H.; Hallman, M. Maternal endotoxin-induced preterm birth in mice: fetal responses in toll-like receptors, collectins, and cytokines. Pediatr Res 2008, 63, 280-286, doi:10.1203/PDR.0b013e318163a8b2.

Reviewer 3 Report

biomolecules-787215

Dietary silk peptide inhibits LPS-induced inflammatory responses by modulating Toll-like receptor 4 (TLR4) signaling

Sungwoo Chei, Hyun-Ji Oh, Kippeum Lee, Heegu Jin, Jeong-Yong Lee and Boo-Yong Lee

This manuscript investigates the role of acid-hydrolyzed silk peptide (SP) in regulating inflammatory responses of macrophages in vitro and in vivo. The authors show that pre-treating mice with SP for 7 days inhibits a LPS-driven inflammatory response in vivo, and the SP modulated both systemic and local responses to the LPS. The authors then showed that SP modulates the LPS-TLR4 signaling cascade in both bone marrow-derived macrophages (BMDMs) and the murine macrophage cell line (RAW264.7).

The manuscript is well presented, and the methods and the statistical analysis appear appropriate.

General comments:

It would have been informative if the authors had been more through in their introduction to the previous publications on anti-inflammatory properties of proteins and peptides isolated from silk moth cocoons. Including:

Eur J Biochem. 1997 Jul 15;247(2):614-9

Biochem J. 1999 May 15;340 ( Pt 1)(Pt 1):265-71.

Infect Immun. 1999 Dec;67(12):6445-53.

These papers even discuss the ability of these compounds to bind LPS.

Therefore, the authors need to define the peptide sequence of the SP used not just the composition, so that the reader can compare this work to other named cocoon-derived compounds/peptides/proteins. Also comment if the SP used in this manuscript is known to bind LPS.

Also, the authors need a rationale for using 100 or 750 mg/kg/day treatment with SP, and why 7 days pre-treatment was used.

In addition, the authors did not use a control group of mice that were injected with either 100 or 750 mg/kg/day SP but not given LPS. What do these doses of SP do to mice in the absence of LPS?

This question also needs to be addressed with the in vitro experiments with either BMDM or RAW cells.

What do these doses of SP do to cells in the absence of LPS?

These are standard experimental conditions that should have been presented.

Also, although the authors show that SP modulates the protein levels of TLR4, what does SP do to a control cellular membrane protein? Is TLR2 affected? What about CD14, or MD-2?

Also this author is not totally convinced the RAW cell data adds much to the manuscript. If the authors had used transfection or knock-down technology to show a specific receptor or signaling molecule was necessary for the SP effect; that would have been beneficial.

Specific comments

Figure 1: The analysis of COX-2 in Fig 1C western blot shows in control samples 2 of 3 have similar levels of COX-2, but the third replicate has no COX-2 signal but all three replicates have good GAPDH signal, yet the quantification on the right hand side of the graph has very small error bars. How can this be true? Please explain. This looks inappropriate. Please also re-check all the western blots from all figures for this error.

Figures 1D, 4B, 4E, 6B. Please do not use “relative cytokine level”. These are data generated with ELISA assays and therefore have actual cytokine concentrations. These values must be used. It is the only way the reader can compare the author’s data with other previously published data. Use actual data values if possible.

Figure 2A. For panel 4 use the same axis convention as the first 3 panels and have SSC-H on the y-axis and mAb staining on the x-axis, it is easier for the reader to comprehend.

Also for Figure 2B, could the authors also show cytospins stained with mAb to specifically define the monocyte/macrophage cells compared to other CD45 positive and negative cells? This would indicate if SP is affecting other cells in the peritoneal cavity.

Minor comments

As the SP is generated from acid hydrolysis can the authors comment on the pH of the material used in the animal and cell culture experiments.

Methods, line 184, use xg not rpm for centrifugation details.

In the manuscript the authors use both euthanized and sacrificed. Keep it standard, use euthanized, euthanasia.

Line 29, reference 3 is added multiple times and is absent from the reference list.

Line 252, title not in bold.

Figure 2B, no scale bars used. Please add.

Author Response

Reviewer3

This manuscript investigates the role of acid-hydrolyzed silk peptide (SP) in regulating inflammatory responses of macrophages in vitro and in vivo. The authors show that pre-treating mice with SP for 7 days inhibits a LPS-driven inflammatory response in vivo, and the SP modulated both systemic and local responses to the LPS. The authors then showed that SP modulates the LPS-TLR4 signaling cascade in both bone marrow-derived macrophages (BMDMs) and the murine macrophage cell line (RAW264.7).

The manuscript is well presented, and the methods and the statistical analysis appear appropriate.

General comments:

It would have been informative if the authors had been more through in their introduction to the previous publications on anti-inflammatory properties of proteins and peptides isolated from silk moth cocoons. Including:

  • We appreciate your constructive comments. We added refences in introduction.

These papers even discuss the ability of these compounds to bind LPS.

Therefore, the authors need to define the peptide sequence of the SP used not just the composition, so that the reader can compare this work to other named cocoon-derived compounds/peptides/proteins. Also comment if the SP used in this manuscript is known to bind LPS.

  • We appreciate your constructive comments. Silk cocoon has peptide sequence; however, it is difficult to define the peptide sequence of SP because SP used in our study was acid hydrolyzed to facilitate ingestion. The data shown in below is mass spectrometry analysis of SP, and it shows that tandem mass spectrum generated by SP components (This data is in our previous paper (in press), and SP used in both papers is the same). According to data, the mean molecular weight of the components of SP was <500 Da. For the same reason, adding to the technical problems, we couldn’t examine the ability of SP to bind LPS.

Also, the authors need a rationale for using 100 or 750 mg/kg/day treatment with SP, and why 7 days pre-treatment was used.

  • Thanks for your comment. We examined the effect of SP on innate immune system in LPS-induced C57BL/6 mice. To investigate the anti-inflammatory effect of SP in the innate immune system by administering the SP for a short period of time, we tried 3 or 5 days of pre-treatment, but the effect was not clear. But when we pre-treated SP for 7 days, it worked, and we used it.

In addition, the authors did not use a control group of mice that were injected with either 100 or 750 mg/kg/day SP but not given LPS. What do these doses of SP do to mice in the absence of LPS?

This question also needs to be addressed with the in vitro experiments with either BMDM or RAW cells.

What do these doses of SP do to cells in the absence of LPS?

These are standard experimental conditions that should have been presented.

  • Your advice is appropriate. In our previous paper[1], we did oral administration of silk peptide with 5-week-old-mouse. As a result, there was no difference in immune cells after five weeks of oral administration of silk peptide. We focused this paper to determine the protective effect of SP response to gram negative bacteria such as LPS.

Reference figure1. Cytokine levels in young (YM, 5-week-old) and old (OM, 14-month-old) C57BL/6 mice serum. (data not published)

Also, although the authors show that SP modulates the protein levels of TLR4, what does SP do to a control cellular membrane protein? Is TLR2 affected? What about CD14, or MD-2?

  • To study the effect of SP on the innate immune system, we used LPS, which is the most commonly used as stimulant in the innate immune analysis. As documented in introduction, LPS is a PAMP found in the outer membrane of Gram-negative bacteria and it binds to Toll-like receptor 4(TLR4) expressed on the surface of immune cells. So, we focused on TLR4 activation induced by LPS rather than other TLRs, and SP showed an outstanding effect in suppressing LPS-induced inflammation through TLR4 signaling. Unfortunately, we couldn’t analyze the impact of the SP on CD14 or MD-2 due to financial problem. So instead, we examined other key members of the TLR4 signaling pathway present in our lab.

Also this author is not totally convinced the RAW cell data adds much to the manuscript. If the authors had used transfection or knock-down technology to show a specific receptor or signaling molecule was necessary for the SP effect; that would have been beneficial.

  • Thanks for your constructive comment. You’re right. It would be much better to use transfection or knock-down technology to prove our hypothesis. However, we could not perform these experiments due to technical problem. Thereby, we confirmed our hypothesis through both experiments: in vitro and in vivo.

Specific comments

Figure 1: The analysis of COX-2 in Fig 1C western blot shows in control samples 2 of 3 have similar levels of COX-2, but the third replicate has no COX-2 signal but all three replicates have good GAPDH signal, yet the quantification on the right hand side of the graph has very small error bars. How can this be true? Please explain. This looks inappropriate. Please also re-check all the western blots from all figures for this error.

  • Thanks for pointing out our mistake. As your kind advice, it is now edited for proper error bars in figure 1C.

Figures 1D, 4B, 4E, 6B. Please do not use “relative cytokine level”. These are data generated with ELISA assays and therefore have actual cytokine concentrations. These values must be used. It is the only way the reader can compare the author’s data with other previously published data. Use actual data values if possible.

  • Thanks for your constructive suggestion. However, we determined that using “relative cytokine level” to show the value of the fold change relative to the control group could better represent the impact of SP on cytokine release. We also focused on examining relative changes upon SP treatment rather than absolute values.

Figure 2A. For panel 4 use the same axis convention as the first 3 panels and have SSC-H on the y-axis and mAb staining on the x-axis, it is easier for the reader to comprehend.

  • According to your kind comment, we edited Figure 2A to make it easier for readers to understand.

Also for Figure 2B, could the authors also show cytospins stained with mAb to specifically define the monocyte/macrophage cells compared to other CD45 positive and negative cells? This would indicate if SP is affecting other cells in the peritoneal cavity.

  • Thanks for your suggestion, but we stained with all alive peritoneal cells to investigate the overall cell morphology. Thus, in Figure 2A, we showed the macrophages through defined morphology, instead of CD45, among the total cells and the number of macrophages were counted. The results suggest that SP reverses the LPS-induced increase in the number of peritoneal macrophages, without affecting the total number of cells.

Minor comments

As the SP is generated from acid hydrolysis can the authors comment on the pH of the material used in the animal and cell culture experiments.

  • The “at pH 7.2” is added in part of Materials and Methods in line 90.

Methods, line 184, use xg not rpm for centrifugation details.

  • All ‘rpm’ written in Materials and Methods are modified to ‘xg’ in line 125,126,152,183, and 185.

In the manuscript the authors use both euthanized and sacrificed. Keep it standard, use euthanized, euthanasia.

  • All ‘sacrificed’ are revised to ‘euthanized’ in line 181 and 216, and Figure 1A.

Line 29, reference 3 is added multiple times and is absent from the reference list.

  • It is now edited.

Line 252, title not in bold.

  • It was our mistake, and now it’s edited.

Figure 2B, no scale bars used. Please add.

  • As your comment, we added scale bars in Figure 2B.

References

  1. Sungwoo Chei, Hyun-Ji Oh, Kippeum Lee, Heegu Jin, Jeong-yong Leeand Boo-Yong Lee. Dysfunction of B Cell Leading to Failure of Immunoglobulin Response is Ameliorated by Dietary Silk Peptide in Aged C57BL6 Mice. FASEB (in press _ under review)

Round 2

Reviewer 3 Report

this reviewer appreciates the difficulties that researchers have regarding ability to provide data with additional experiments.

The other issues have been resolved

Author Response

Dear Reviwer 

We are very appreciate to have been given the opportunity to revise our manuscript. We revised our manuscript.

We hope that the manuscript is now acceptable for publication in Biomolecules.

Sincerely, 
Boo-Yong Lee